# Identifying Conserved Generic *Aspergillus* spp. Co-Expressed Gene Modules Associated with Germination Using Cross-Platform and Cross-Species Transcriptomics

**DOI:** 10.3390/jof7040270

**Published:** 2021-04-01

**Authors:** Tim J. H. Baltussen, Jordy P. M. Coolen, Paul E. Verweij, Jan Dijksterhuis, Willem J. G. Melchers

**Affiliations:** 1Department of Medical Microbiology, Radboud Institute for Molecular Life Sciences, Radboud University Medical Center, 6525 GA Nijmegen, The Netherlands; Jordy.Coolen@radboudumc.nl (J.P.M.C.); Paul.Verweij@radboudumc.nl (P.E.V.); Willem.Melchers@radboudumc.nl (W.J.G.M.); 2Center of Expertise in Mycology Radboudumc/CWZ, 6532 SZ Nijmegen, The Netherlands; 3Westerdijk Fungal Biodiversity Institute, 3584 CT Utrecht, The Netherlands

**Keywords:** aspergillus, germination, consensusWGCNA, isotropic growth, polarized growth, RNA-Seq, microarray

## Abstract

*Aspergillus* spp. is an opportunistic human pathogen that may cause a spectrum of pulmonary diseases. In order to establish infection, inhaled conidia must germinate, whereby they break dormancy, start to swell, and initiate a highly polarized growth process. To identify critical biological processes during germination, we performed a cross-platform, cross-species comparative analysis of germinating *A. fumigatus* and *A. niger* conidia using transcriptional data from published RNA-Seq and Affymetrix studies. A consensus co-expression network analysis identified four gene modules associated with stages of germination. These modules showed numerous shared biological processes between *A. niger* and *A. fumigatus* during conidial germination. Specifically, the *turquoise* module was enriched with secondary metabolism, the *black* module was highly enriched with protein synthesis, the *darkgreen* module was enriched with protein fate, and the *blue* module was highly enriched with polarized growth. More specifically, enriched functional categories identified in the *blue* module were vesicle formation, vesicular transport, tubulin dependent transport, actin-dependent transport, exocytosis, and endocytosis. Genes important for these biological processes showed similar expression patterns in *A. fumigatus* and *A. niger*, therefore, they could be potential antifungal targets. Through cross-platform, cross-species comparative analysis, we were able to identify biologically meaningful modules shared by *A. fumigatus* and *A. niger*, which underscores the potential of this approach.

## 1. Introduction

The genus *Aspergillus* consists of at least 450 species [1] that occur worldwide and are members of all habitats. They grow in soil, are associated with plants, and even colonize oceans. The genus coexists with other living organisms, and this association regularly develops into an (opportunistic) pathogenicity, for example *Aspergillus sydowii* on coral reefs and *Aspergillus niger* on onions [2,3]. One of the important means of distribution within the genus are single-celled survival structures, called conidia, that are released into the air. Conidia of fungal species in different genera, including *Aspergillus*, are globally distributed [4], and, due to their small size, can enter the lungs of animals, including humans [5]. These conidia contain a rodlet layer that effectively shields them from the human immune system, but rarely causes infections [6,7]. Nevertheless, with Aspergilli that can grow at hypoxic conditions and body temperatures, and with a person with immune deficiencies, a serious risk for infection develops. In the case of *A. fumigatus*, and to a lesser extent the species *A. flavus, A. niger, A. terreus,* and *A. nidulans,* this can lead to severe lung disease, including chronic pulmonary aspergillosis (CPA) and invasive aspergillosis (IA).

CPA is estimated to affect 3 million people worldwide, and more than 200,000 people are affected by IA each year [8,9]. The overall mortality rate of IA is 50% even if diagnosed and treated, but if diagnosis is delayed or missed, or if infection is caused by an antifungal-resistant strain, then mortality may increase to nearly 100% [8]. Although IA is traditionally observed in patients with neutropenia, over the past few decades, increasingly non-neutropenic hosts are at risk of developing IA, including patients with severe influenza and coronavirus disease (COVID-19) [10,11]. The pathogenesis of IA is characterized by the germination of inhaled conidia, and, if not internalized by epithelial cells and phagocytized by alveolar macrophages, hyphal growth and penetration into lung tissue [5,12].

Germination of inhaled conidia is crucial to establish an infection in the host. Although dormant conidia may be immunologically inert, once germinating, they are able to express a repertoire of factors that enable them to develop and grow in lung tissue [5]. After a conidium breaks dormancy, the germination is characterized by two stages: isotropic growth and polarized growth. Dormant conidia are protected by intra- and extracellular molecules, among them proteins, against unfavorable conditions, such as oxidative stress, variations in pH, osmotic pressure, dehydration, thermal stress, and UV radiation [13]. Breaking of dormancy in *Aspergillus* conidia initiates the transition of a resting cell into a vegetative active cell, and, in a number of cases, transition is the result of the presence of nutrients in the environment, such as sugars and amino acids [14,15,16,17]. Breaking of dormancy includes a massive reorganization of the transcriptome [18,19] and degradation of compatible solutes [20], and is accompanied by an increase of the cellular perimeter dubbed isotropic growth (“swelling”) [21]. Isotropic swelling of the conidia is followed by polarized growth. In this phase, the growth is directed to a local region on the cell surface [22], which results in the formation of a germ tube [18,22,23,24,25,26,27].

Transcriptomic and proteomic studies over the past two decades have provided important information on genes and proteins involved in germination [18,19,23,24,25,26,28,29,30,31,32,33,34,35,36]. Remodelling of the cell wall, together with metabolic activities required for cellular growth, such as protein synthesis and carbohydrate metabolism, are important during isotropic growth. These metabolic activities, together with the functional organization of the hyphal tip and the cell cycle machinery, are important during polarized growth. A more profound understanding of the transitions from dormant conidia via germ tube initiation to hyphal tip formation may be vital in the search for possible novel strategies to eradicate early infection.

In this study, we performed a cross-platform, cross-species comparative analysis of germinating *A. fumigatus* and *A. niger* conidia using transcriptional data from two published studies [18,23]. The different transcriptional datasets were integrated into one, and a consensus co-expression network analysis was applied to identify common biological processes during germination.

## 2. Materials and Methods

### 2.1. Transcriptomic Data

In this study, published transcriptomic data of germinating *A. fumigatus* and *A. niger* conidia were used [18,23]. The raw data were accessed through the NCBI Gene Expression Omnibus (GEO) accession number GSE36439 (GEO. Available online: www.ncbi.nlm.nih.gov/geo/ accessed on 24/03/2020) and the NCBI Sequence Read Archive (SRA) accession number PRJNA408076 (SRA. Available online: https://www.ncbi.nlm.nih.gov/sra accessed on 24/03/2020). The *A. fumigatus* data were obtained via Illumina RNA-Seq and the *A. niger* data via Affymetrix *A. niger* Genome Genechips. The *A. niger* strain N402 [37] and *A. fumigatus* strains AfIR974 and AfIR964 were used in the studies.

### 2.2. Orthology Inference Using Reciprocal Best Hits (RBH) Method

To compare the transcriptomic profiles of *A. fumigatus* and *A. niger*, pairs of genes were identified in two different genomes using the RBH method. This method entails that the pairs of genes between two species are more similar to each other than to any other gene in the other genome [38]. NCBI’s BLAST (version 2.10.1+) was first used to create two databases of the protein sequences of *A. fumigatus* af293 [39] and *A. niger* CBS513.88 [40]. The *A. fumigatus* annotated protein sequences were downloaded from Ensembl Fungi (available online: http://ftp.ebi.ac.uk/ensemblgenomes/pub/release-30/fungi/fasta/aspergillus_fumigatus/, accessed on 06/10/2020 (Aspergillus_fumigatus.CADRE.30.pep.all.fa)). The *A. niger* annotated protein sequences were downloaded from Aspergillus Genome Database (available online: http://www.aspgd.org/, accessed on 06/10/2020 (A_niger_CBS_513_88_orf_trans_all.20110819.fasta)). The specific command lines used to build the protein sequence databases of *A. fumigatus* and *A. niger* are presented in Appendix A.

A BLASTp all vs. all was performed using the *A. fumigatus* protein sequences as the query and the *A. niger* protein database as the subject, and vice versa. The command lines used for this query protein to subject protein comparison are presented in Appendix A. The additional options for blastp were a final Smith–Waterman alignment and a maximum e-value threshold of 1 × 10^−6^ [41]. Additional requirements were a query coverage per subject of 60%, a minimum bit score of 80, and a minimum percent identify of 30. For selecting the RBH hits results, the query protein to subject protein comparisons were first sorted from lowest to highest e-values, then from highest to lowest bit scores. After sorting the results, the first hit for each query was therefore the best hit. Finally, each first hit in the first direction was compared with the first hit for each query in the opposite direction. Using the RBH method, we identified 6,598 orthologous genes between *A. fumigatus* and *A. niger*.

### 2.3. Data Integration and Exploratory Analysis

The gene pairs were used to integrate the RNA-Seq dataset with the microarray dataset. The datasets of *A. fumigatus* and *A. niger* contained 19 and 15 samples, respectively. In the first step, the merge function from the base R package was used to combine both sets based on the identified gene pairs [42]. The integrated dataset was log transformed using the log1p function from the R base package, which computes log_e_(1 + x) [42]. A principal component analysis was performed on the log transformed data. The pca and biplot functions from the R package PCA tools were used to generate the principal components and corresponding plots [43]. Next, normalization of the integrated dataset was applied using the NormalizeBetweenArrays function from the limma R package [44], as previously proposed by Castillo et al. [45]. The R package ggplot2 was used to plot the data before and after normalization [46].

### 2.4. Consensus Weighted Gene Co-Expression Network Analysis (consensusWGCNA)

The integrated dataset was analyzed using a constructed consensus network. The network was built using the WGCNA R package [47,48]. The integrated dataset was put into a multi-set format suitable for consensus analysis. The trait data were matched with the expression samples for which they were measured. The corresponding traits were dormant (0 h), isotropic growth (2 h, 4 h), and polarized growth (6 h, 8 h). Construction of the weighted gene network entails the choice of a soft thresholding power β. The power β of 13 (R^2^ of 0.82) was chosen based on the criterion of approximate scale-free topology [49]. The function blockwiseConsensusModules was used for network construction and consensus module detection. Other thresholds included a minimum module size of 30, a cut height for the merging of modules of 0.30 (modules whose eigengenes were correlated above 0.7 were merged), correlation option Pearson (corType), adjacency function option (networkType) signed hybrid, and a topological overlap option (TOMType) signed. The soft power β was used to calculate the Pearson’s correlation between all genes. To minimize the effects of noise and spurious associations, the adjacency matrix was converted to a consensus topological overlap measure (TOM). For calculation of the consensusTOM, individual TOMs were scaled by full quantile normalization (networkCalibration = full quantile). The modules detected by the blockwiseConsensusModules function were assigned a random color. For each module, a module eigengene was calculated by the first principal component. The module eigengene could be regarded as the best representation of the gene expression patterns of that module. Next, trait data and module eigengenes were used to calculate the module–trait relationships by using the Pearson’s correlation between the trait of interest and the module eigengene. To summarize the two sets into one (i.e., for detection of modules with similar correlation to the external traits), we used a conservative method: for each module–trait pair, we took the correlation that had the lower absolute value in the two sets if the two correlations had the same sign, and zero relationship if the two correlations had opposite signs. Modules with a correlation ≥ |0.70| and *p*-value ≤ 0.01 were selected for further analyses.

### 2.5. Functional Classification

Consensus modules that were correlated with any of the growth phases were selected for functional enrichment analysis to understand the biological function of the modules. To analyze the corresponding gene lists, we used the online webtool FungiFun2 (version 2.2.8) and used the functional ontologies from the Functional Catalogue (FunCat) [50,51]. The default settings of the FungiFun2 webtool were used, except for the background; as the background, the 6598 identified orthologous genes were used.

## 3. Results

### 3.1. Orthology Inference, Data Integration, and Exploratory Analysis

The transcriptomic data used in this comparative study were generated by two different transcriptional profiling platforms (i.e., Illumina NextSeq500 and Affymetrix *A. niger* Genome Genechips). For identification of orthologous gene pairs between the two species, the RBH method was used, which resulted in 6598 orthologous gene pairs. Next, the identified orthologs were used to integrate the datasets from both RNA-Seq and microarray technologies. The first two principal components are plotted in Figure 1 to visualize the similarities and dissimilarities between the samples. The integrated dataset was log transformed using the log1p function to avoid the variance measure being dominated by highly expressed, highly variable genes [42,52]. Variation between the two species was explained by the first principal component, whereas variation between the different time points was explained by the second and third principal component. Variation in microarray data between germinating *A. niger* conidia (2–8 h) were small, as those samples were clustered together on PC2. Only the 0 h samples of *A. niger* were substantially different from all other time points. Larger variations were observed between *A. fumigatus* RNA-Seq samples, with dissimilarities between the 2–4 h samples and 6–8 h samples. The 0 h samples were substantially different from the 6–8 h samples. The third principal component was plotted to better explore the variation between the samples of *A niger*. Variation between *A. niger* samples was explained by PC3 rather than PC2, whereas variation between *A. fumigatus* samples was explained by PC2 and PC3.

The raw data, i.e., RNA-Seq counts and microarray fluorescence intensities, are plotted in Figure 2A to show the difference of the dynamic range between the datasets. A larger dynamic range of the RNA-Seq samples was observed compared with microarray samples. Figure 2B shows the results of the joint normalization, where the dynamic range between the samples has been corrected. The outliers were left out of Figure 2 for better visualization of the data. Normalized data including outliers are plotted in Appendix A.

### 3.2. Consensus Co-Expression Network Analysis

To examine the transcriptomic similarities between germinating *A. fumigatus* and *A. niger* conidia, we constructed a consensus gene co-expression network. Co-expression networks constructed from gene expression data suggest functional relationships between genes [49,53]. Consensus modules may contain shared biological pathways between the compared datasets. The consensusWGCNA detected 25 highly co-expressed gene modules that varied greatly in size (41–992 genes). Each module was labelled by a color, and, henceforth, we will refer to each module by its corresponding color.

Next, we used the module eigengenes to relate the consensus modules to external sample information. An eigengene is the first principal component of that module, and may be regarded as a representative of the gene expression patterns in the corresponding module. The external trait information was matched with the expression samples for which they were measured. The defined traits were dormant (0 h), isotropic growth (2 h, 4 h), and polarized growth (6 h, 8 h). Each gene was assigned to a single module, but each module had two consensus module eigengenes. This was because each orthologous gene had a particular expression pattern in *A. fumigatus* and a different expression pattern in *A. niger*. To determine if any of the 25 modules were associated with the traits, we calculated the correlation of the module eigengenes with each trait for *A. fumigatus* and *A. niger* (Figure 3A,B). To identify the modules that were highly correlated to any of the traits in both species (consensus modules), the two sets were summarized into one: for each module–trait, pair we took the correlation that had the lower absolute value in the two sets if the two correlations had the same sign, and zero relationship if the two correlations had opposite signs (Figure 3C). Only modules that had a significant correlation with an external trait are shown in Figure 3A–C. The *turquoise* and *black* modules were highly correlated to the dormant phase (0.89 and −0.72, respectively), *midnightblue* was correlated to isotropic growth (0.82), and the *darkgreen* and *blue* modules were highly correlated to polarized growth (0.73 and 0.79, respectively). For all 25 modules and corresponding correlation and *p*-values to the traits, see Appendix A.

To retrieve the biological function of the highly correlated consensus modules, we performed a functional enrichment analysis using functional ontologies from FunCat [51]. The consensus module *midnightblue* did not show any significant results (*p* > 0.05). The *turquoise* module was found to represent mostly secondary metabolism and fatty acid and carbohydrate metabolism. The *black* module was highly enriched with protein synthesis genes, the *darkgreen* module was enriched with ubiquitin-related genes, and the *blue* module was highly enriched in polarized growth genes. The detailed results are shown in Figure 4 and Figure 5 and Appendix A. The specific biological processes and molecular functions identified in each of the significant modules (i.e., *turquoise, black, darkgreen,* and *blue*) are described in the next section. Closely related co-expression modules may form a biologically meaningful meta-module. In Figure 6, the clustering dendrogram of consensus module eigengenes is plotted for identifying meta-modules. The meta-modules are further described in next section.

### 3.3. Modules

#### 3.3.1. Turquoise

The *turquoise* module contained 992 genes, and was the largest module detected by the consensusWGCNA. The gene expression patterns in this module showed that transcripts were high in dormant conidia, and then decreased during isotropic and polarized growth (Figure 7A). The module was mostly enriched with genes involved in metabolism (secondary metabolism, fatty acid and carbohydrate metabolism), but other FunCat categories were also enriched, such as transcription and cellular transport (Figure 6). The *darkgrey* module was closely related to the *turquoise* module, and contained 41 genes. However, the FunCat enrichment did not show any significant hits (*p* > 0.05).

#### 3.3.2. Black

The *black* module contained 308 genes that were mostly involved in protein synthesis (12), but categories associated with energy (02) were also enriched (Figure 4). Highly enriched categories were electron transport and membrane-associated energy conversion (02.11), ribosome biogenesis (12.01), ribosomal biogenesis (12.01.01), aminoacyl-tRNA-synthases (12.10), protein binding (16.01), and electron transport (20.01.15). Conidial outgrowth involves a fermentative metabolism, followed by a switch to respiration [33]. Concomitant with respiratory metabolism during the breaking of dormancy is protein synthesis, which is one of the earliest measurable biochemical changes during germination [54]. Transcriptomic and proteomic analyses have shown that the breaking of dormancy is characterized by an immediate onset of protein synthesis [19,25,30]. The transition from dormant conidia to isotropically expanding conidia and eventually germ tube formation involves biosynthetic machineries for which protein synthesis is required [13].

The gene expression patterns in the module were slightly different between *A. fumigatus* and *A. niger* (Figure 7B). In *A. fumigatus*, transcripts increased after 2 h, then decreased slightly after 4 h. This pattern was also observed during polarized growth; after 6 h, transcripts increased, then decreased slightly after 8 h. In *A. niger*, transcripts were low during the dormant phase, then increased and remained high during isotropic and polarized growth. These differences during isotropic and polarized growth between *A. fumigatus* and *A. niger* can also be seen in Figure 3, as the correlation scores are low.

The *brown* module was closely related to the *black* module, and contained 658 genes. The FunCat analysis mainly showed enriched categories associated with transcription (11) and protein synthesis (12), which were connected to the categories enriched in the *black* module (Appendix A). Identical FunCat categories identified were ribosome biogenesis (12.01), ribosomal proteins (12.01.01), and translation (12.04). Various categories were associated with transcription (11), such as RNA synthesis (rRNA (11.02.01), tRNA (11.02.02) and mRNA (11.02.03)), RNA processing (rRNA (11.04.01), tRNA (11.04.02), and mRNA (11.04.03)), and RNA modification (rRNA (11.06.01) and tRNA (11.06.02)).

#### 3.3.3. Darkgreen

The *darkgreen* module contained 65 genes, and was the second smallest module detected by the consensusWGCNA. The gene expression pattern in this module increased during isotropic and polarized growth (Figure 7C). Functional enrichment showed that only two categories were enriched in this module (Figure 4). The categories were conjunction of sulfate (01.02.03.04) associated with metabolism (01) and modification by ubiquitin-related proteins (14.07.07) associated with protein fate (14). The *magenta* module was closely related to the *darkgreen* and *blue* modules. This module contained 271 genes, and the FunCat analysis showed only three categories enriched: proteasomal degradation (14.13.01.01) and protein processing (14.07.11) associated with protein fate (14) and degradation of leucine (01.01.11.04.02) associated with metabolism (01).

#### 3.3.4. Blue

The *blue* module contained 770 genes, and was the second largest module detected by the consensusWGCNA. This module was highly enriched in genes involved in polarized growth, together with genes involved in cell cycle and DNA processing (10) (Figure 7D, Figure 5).

##### Mitosis and Septum Formation

Highly enriched categories were mitotic cell cycle and cell cycle control (10.03.01), mitotic cell cycle (10.03.01.01), M phase (10.03.01.01.11), cytokinesis (cell division)/septum formation and hydrolysis (10.03.03), and nuclear migration (10.03.04.09). In *A. fumigatus*, the first mitosis was completed in 22% of the cells before polarized growth with the formation of a germ tube was initiated [27]. The number of nuclei increases during isotropic growth, and continues to increase during polarized growth [16]. Nuclear division is followed by the migration of nuclei into the elongating germ tube and septum formation. The septin *aspC* was found in the *blue* module, and plays a role in normal development and morphogenesis [55]. In *∆aspC* strains, germ tubes emerged early, and multiple germ tubes, together with early branching, were observed.

##### Protein Processing

Other highly enriched categories were protein targeting, sorting and translocation (14.04), protein processing (14.07.11), assembly of protein complexes (14.10), protein binding (16.01), structural protein binding (16.07), and enzymatic activity regulation/enzyme regulator (18.02.01).

##### Onset of Polarization

Highly enriched categories important for germ tube elongation were protein transport (20.01.10), vesicular transport (20.09.07), vesicle formation (20.09.07.25), cytoskeleton-dependent transport (20.09.14), cell growth/morphogenesis (40.01), cytoskeleton/structural proteins (42.04), intracellular transport vesicles (42.09), and budding, cell polarity, and filament formation (43.01.03.05). Polarized growth is characterized by the restriction of expansion of the cell wall on the swollen spore, which leads to a tubular outgrowth, the germ tube. Before this, a position on the plasma membrane has to be confined [22] for localized vesicle fusion and membrane extension. This will include orchestration of the cytoskeleton to traffic and deliver vesicles, which will lead to localized membrane expansion and cell wall deposition. This extension is the first appearance of germination, and the bulge will expand to a small tube, the germ tube, which extends by tip growth. During this stage, a septum is delineated at the base of the germ tube, and nuclei are transported into the growing cell. Several studies have identified a marked increase in the growth speed of the germ tube that will branch. In many cases, a second germ tube is formed on the swollen spore. Hyphae that grow at a higher velocity possess a so-called vesicle supply center [56], also designated as the Spitzenkörper (SPK) [57,58]. This is a dynamic structure containing different types of vesicles and cytoskeletal elements that maintain polarized growth. At early germination, similar organization is expected, but operating in a more diffuse way [59].

All of these processes were confirmed in the FunCat analysis, which showed categories growth/morphogenesis (40.01) and directional cell growth (morphogenesis) (40.01.03) enriched in the *blue* module. Additionally, the categories cytoskeleton/structural proteins (42.04), actin cytoskeleton (42.04.03), microtubule cytoskeleton (42.04.05), and bud/growth tip (42.29) were enriched. Microtubules are primarily responsible for the transport of secretory vesicles to the location of localized expansion and in fully growing hyphae, the SPK, while actin filaments primarily control the organization of vesicles and facilitate transport/delivery to the plasma membrane [60]. Enriched FunCat categories associated with transport were protein transport (20.01.10), vesicular transport (20.09.07), vesicle formation (20.09.07.25), tubulin-dependent transport (20.09.14.01), actin-dependent transport (20.09.14.02), exocytosis (20.09.16.09.03), and receptor-mediated endocytosis (20.09.18.09.01), among others (Figure 7). Several genes encoding secretion-related GTPases and interacting proteins were present in the *blue* module, such as An01g04040/Afu1g04940, An01g06060/Afu1g02190, An08g03690/Afu1g11730, An14g00010/Afu4g04810, and An18g02490/Afu5g11900 [40] (Table 1). In *A. nidulans*, homologs of An14g00010/Afu4g04810 (RabD) and An01g06060/Afu1g02190 (RabE) were detected in the SPK [61]. Microtubule and actin-dependent transport involves the trafficking and delivery of vesicles to the plasma membrane, which will lead to localized expansion of the cell membrane. Other proteins are also involved in the distribution of vesicles to their final destination.

In *S. cerevisiae*, members of the Cdc42 complex are Cdc42, Cdc24, Bem1, Cla4, and Ste12. In the *blue* module, homologs of the Rho-type GTPase *cdc42* (*cftA)* and its guanidine exchange factor (GEF) *cdc24* (An04g05150) were present, together with Rho-type GTPase *racA* (*racA*) (Table 1). These homologues have been studied in *A. nidulans* (*modA* and *racA*) and *Neurospora crassa* (*CDC-42* and *RAC-1*) [62,63]. In *A. nidulans*, ModA (Cdc42) and RacA (Rac1) share an overlapping function required for polarity establishment. The double knockout *∆cdc42∆rac1* appeared to be synthetically lethal. Additionally, GEF Cdc42 was required for the establishment of hyphal polarity, and localized to hyphal tips [64]. In *N. crassa,* the spatial distribution of the two Rho-type GTPases Cdc42 and Rac1 changes during the various differentiation stages. Before the breakage of symmetry in conidia, the localization and localized activation of Cdc42 and its GEF Cdc24 occur. After emergence of the germ tube, Rac1 is recruited at the developing tip. Together, Cdc42 and Rac1 regulate the negative chemotropism displayed during germ tube formation. 

##### Establishment of Polarized Growth, Hyphal Elongation

Cell wall expansion is necessary for every expansion of the cell, being isotropic or polarized. Cell wall biosynthesis genes were also identified, belonging to different glycoside hydrolase and glycoside transferase families, such as GH13 (An09g03100/Afu3g00900), GH16 (An07g07530/Afu2g03120, An16g02850/Afu3g09250), GH17 (An16g07040/Afu8g05610), GH18 (An01g05360/Afu1g02800), GH72 (An03g06220/Afu8g02130, An08g07350/Afu6g11390), and GT2 (An09g02290/Afu8g05630) (CAZy. Available online: http://www.cazy.org/, accessed on 18/03/2021). Another identified cell wall biosynthesis gene (An05g00130/Afu2g07590) has a predicted role in the (1->6)-beta-D-glucan biosynthetic process. Some of the cell wall biosynthesis genes are glycosylphosphatidylinositol (GPI) anchored enzymes, and one is a chitin synthase. Chitin synthases and glucan synthase complexes are transported in vesicles and fuse with the plasma membrane, where the enzymes are inserted to synthesize chitin and glucan polysaccharides [65]. The glycoproteins are also transported through the secretory pathway to the growth tip and, after exocytosis, remain attached to the plasma membrane by the GPI anchor.

The SPK is vital for polarity maintenance during hyphal tip extension. The polarity maintenance machinery consists of cytoskeleton components, such as microtubules and actin filaments, and several groups of proteins termed the Cdc42 complex, polarisome, and Arp2/3 complex. These complexes are located in the growing tip area close to the apical plasma membrane [66]. Microtubules regulate the position of proteins, such as cell end markers. The cell end marker *teaR* (An18g04780) was found in the *blue* module. In *A. nidulans,* TeaR is anchored to the plasma membrane, and directly interacts with TeaA, another cell end marker [67]. This interaction at the apical membrane is important for the recruitment of additional downstream components, including the formin SepA, which is involved in the polymerization of actin filaments for targeted vesicle transport [68]. 

Components of the polarisome act downstream of the Cdc42 complex, and are conserved from yeast to filamentous fungi [69]. Only one of three components of the *A. niger* polarisome was found in the *blue* module, *spaA* (Table 1). SpaA localizes exclusively at the hyphal tip and plays a role in polarity maintenance [70]. The Arp2/3 complex is another group of proteins involved in polarity maintenance, endocytosis, and actin polymerization. The complex includes Arp2, Arp3, Arc40, Arc35, Arc18, Arc19, and Arc15 in *S. cerevisiae* [71]. Several Arp2/3 homologs were identified in the *blue* module, such as Arp2 (An08g06400), Arc35 (An01g05510), Arc19 (An12g08380), and Arc18 (An16g01570) (Table 1).

Vesicles transport cell wall-modifying enzymes, substrates, and the cell membrane required for expansion to the growing tip. The exocyst is a protein complex involved in vesicle docking and fusion to the plasma membrane, and was originally identified in the budding yeast *S. cerevisiae* [72,73]. This complex consists of eight proteins, Sec3, Sec5, Sec6, Sec8, Sec10, Sec15, Exo70, and Exo84, and interacts with Rho-type GTPases Cdc42, Rho1, and Rho3, as well as with Rab GTPase Sec4, which is present on the membrane surface of vesicles [72]. In the *blue* module, we identified homologs of Sec5 (An08g05570/Afu1g12790), Exo84 (An08g07370/Afu6g11370), Sec4 (An14g00010/Afu4g04810), and Rho1 (An18g05980/Afu6g06900) (Table 1). In *N. crassa*, a mutation in the Sec5 homolog resulted in swollen conidia and altered hyphal growth, indicating its role in polarity establishment and maintenance [74]. The Rab GTPase SrgA (Sec4) was involved in vesicle secretion and filamentous growth in *A. niger* [75,76]. Another group of proteins that facilitate vesicle docking and fusion to the plasma membrane are soluble N-ethylmaleimide-sensitive fusion protein (NSF) attachment protein receptors (SNAREs). At the hyphal tip, SNAREs present on the target membrane (t-SNAREs) pair with SNAREs present on the vesicles (v-SNAREs) to mediated the fusion of membranes [77]. Several SNAREs and SNARE interacting genes were identified in the *blue* module, such as An02g01580/Afu2g12870, An04g07020/Afu4g10040, An07g02170/Afu7g05735, An07g09960/Afu1g07420, and An15g01380/Afu6g04150 [40] (Table 1). Endocytosis is the reverse process, characterized by the formation of membrane vesicles that are invaginated and included in the vesicle transport routings. It occurs in germinating spores [78,79] at the onset of polarization. In the case of growing hyphae, endocytotic vesicles are internalized most strongly in a collar-like region behind the hyphal apex, and fuse with early endosomes [80], which participate in tip growth.

## 4. Discussion

The process of germination of conidia involves the transition from a dormant, stress-resistant cell with low metabolic activity into a vegetatively active fungal hypha. In this study, the transcriptomic changes are studied throughout this transition. In this cross-platform, cross-species comparative analysis, we studied conidial germination of two *Aspergillus* species: *A. niger* and *A. fumigatus*, enabling us to integrate the transcriptional expression of two related species and two different techniques and providing biological insights in germination of conidia. 

Firstly, to perform this cross-platform, cross-species comparative analysis, we used the following bioinformatic approach: (i) a selection of 6598 ortholog genes was necessary to integrate the two datasets, which enclosed nearly 50% of the *A. niger* genes and over 50% of the *A. fumigatus* genes. A comparison of 34 ascomycete genomes, including 19 *Aspergillus* species, showed that ~8500 genes were pan-fungal, which was inferred from MCL clustering of proteins [81]. For our analysis, the two datasets needed to be integrated one to one based on orthologous genes. Finding best hits using the RBH method involved sorting the results from lowest to highest e-values, then, from highest to lowest bit-scores. The first hit within the sorting would be the best hit. If the next best hit had the same bit-score and e-value, there would be more than one best hit (co-orthologs). In our study, the co-orthologs were discarded, which may be the cause of the difference in identified orthologous genes, together with the different methods used in both studies. (ii) Normalization of the intensities was done, as both techniques have different expression values, i.e., RNA-Seq counts and microarray fluorescence intensities. (iii) Expression patterns during germination stages were compared. Presently, RNA-Seq has emerged as the technology of choice for gene expression profiling [82]. RNA-Seq is able to detect novel transcripts, map exon/intron boundaries (if full genomes are available), and reveal splice variants. Additionally, RNA-Seq provides more resolution to detect extreme expression values, such as genes with very low transcript counts and genes with extremely high transcript counts [83]. However, microarrays are reliable, and the variety of datasets publicly available offers an exceptional opportunity to perform cost-effective and insightful comparisons.

Secondly, the cross-platform, cross-species analysis confirmed the occurrence of conserved, generic, and functionally important biological processes during germination, which are independent of a single technology. This is more interesting, as *A. fumigatus* belongs to subgenus *Fumigati*, section *Fumigati* and *A. niger* to subgenus *Circumdati,* section *Nigri* [1]. A phylogenetic analysis showed that section *Nigri* is more closely related to subgenus *Nidulantes* than to *Circumdati* [84]. However, based on phenotypic and extrolite data and their phylogenetic analysis, Houbraken et al. [1] maintained section *Nigri* in subgenus *Circumdati* until more data supporting the analysis of Steenwyk et al. [84] become available. Additionally, experimental conditions of both studies were different, such as the pre-culture medium and germination medium. Nutritional environment during sporulation, as well as during germination, affects the rate of the breaking of dormancy and growth in *A. fumigatus* [85]. *A. fumigatus* strains were cultivated on Sabouraud agar slants (dextrose 40 g/L, peptone 10 g/L, agar 20 g/L, pH = 5.6) for five days at 30 °C, and *A. niger* was cultivated on complete medium (CM) (1.5% agar, 6.0 g NaNO3, 1.5 g KH2PO4, 0.5 g KCl, 0.5 g MgSO4, 4.5 g D-glucose, 0.5% casamino acids, 1% yeast extract, and 200 μL trace elements per liter) for 12 days at 25 °C. RNA was extracted after growth in liquid RPMI 1640 (Gibco^®^ life technologies) and liquid CM for *A. fumigatus* and *A. niger,* respectively. Morphological changes during germination, such as swelling and germ tube formation, were observed at similar time points, despite different pre-culture and germination conditions. The bioinformatic analysis was focused on identifying biological similarities associated with germination between *A. fumigatus* and *A. niger.* The different experimental setup, together with differences in RNA extraction and library preparation of the original studies, will doubtlessly cause biological differences. By integrating the expression data one on one based on orthologous gene pairs and constructing a consensus gene co-expression network, we excluded the biological differences and identified biological similarities.

A consensusWGCNA was performed to examine the transcriptomic similarities between germinating *A. fumigatus* and *A. niger* conidia. Recently, numerous studies have been published using WGCNA to identify co-expressed genes related to an external trait in various fields [86,87,88,89,90,91]. In this study, we detected co-expressed consensus modules between *A. fumigatus* and *A. niger* and investigated the module relationships with the different morphological phases in germination. Five co-expression modules associated with either the dormant, isotropic, or polarized phase were identified. Genes within each module were considered to be related to each other in function, or could work cooperatively in specific molecular processes. Functional enrichment on the larger modules, such as the *black* and *blue* modules, adequately showed clustering of genes involved in similar or identical pathways. However, the *midnightblue* module (130 genes) did not show enrichment of a functional category, which could be because of the poorly characterized *A. fumigatus* and *A. niger* genomes. The module contained ~40 genes without annotation and almost all other annotations were putative. Additionally, in both studies, RNA was extracted from a population of germinating conidia. Conidial germination was relatively synchronous when appropriate exogenous nutrients were present [16]. However, in the transition from dormant conidia to swelling to germ tube elongation, conidial swelling is the middle phase, and therefore difficult to characterize when analyzing the transcriptome of a population of conidia.

The gene expression pattern in the *black* module, associated with protein synthesis, showed an increase after 2 h. Lamarre et al. showed similar results in *A. fumigatus;* 30 min post-dormancy transcripts were identified from protein synthesis, carbohydrate metabolism, protein complex assembly, and RNA binding protein [25]. This was recently confirmed by Danion et al., where germination was induced by the presence of nutrients, such as carbon and nitrogen, without the novo RNA transcription [16]. The proteome profile of *A. flavus* at the conidial germination stage resulted in overrepresented protein synthesis categories [26]. The gene expression pattern in the *blue* module, associated with polarized growth, showed a strong increase after 6 h. In several *Aspergillus* spp, germ tube formation was observed after 6–8 h of germination [18,23,24,25,26]. Our co-expression network analysis using only *A. fumigatus* RNA-Seq data also identified a module associated with the onset of polarized growth. However, the consensus co-expression analysis performed in this study identified detailed functional categories associated with the tip growth and formation of an SPK [65,92,93,94]. Oda et al. found that transcript levels were steady for genes encoding the Cdc42 complex and polarisome during the switch from isotropic to polarized growth, such as *modA* (An02g14200), *cdc24* (An04g05150), and *spaA* (An07g08290) [24], whereas our data showed a strong increase of these transcripts in the *blue* module in both *A. fumigatus* and *A. niger*. Besides the modules significantly correlated to a trait, closely related modules, termed meta-modules, were analyzed, as these modules may be biologically similar [53]. The higher-level cluster of the *black* and *brown* module showed a relationship between the gene transcripts in each module. The *black* module was highly enriched with protein synthesis genes, and, similarly, the *brown* module was highly enriched with transcription and protein synthesis genes.

Thirdly, these insights enable us to define processes that might be used as therapeutic targets to suppress antifungal development in the lungs. Antifungals targeting germination processes would only work as prophylaxis, as only established infections consisting of hyphae will be diagnosed and treated. Conidia swelling inside causes the lungs to need to remodel their cell wall, therefore, the enzymatic activity of glycoside hydrolases and glycoside transferases may be potential targets for prophylaxis. Before germ tube formation, a position on the plasma membrane has to be confined for localized vesicle fusion and membrane extension. This ergosterol-enriched cap in germinating conidia at the site of germ tube formation could be an attractive target, as the prevention of this ergosterol patch may disturb the localized transport to the site of polarization. Processes important for germ tube formation and tip elongation, such as vesicle transport and exocytosis/endocytosis, might also be feasible as antifungal development targets.

## 5. Conclusions

In this study, we demonstrated the possibility for comparative analysis between *Aspergillus* spp using two different transcriptional profiling platforms, which introduces the opportunity to perform cost-effective insightful comparisons. The consensus gene co-expression network detected modules associated with transcription and protein synthesis and polarized growth. Through cross-platform, cross-species comparative analysis, we were able to identify biologically meaningful modules shared by *A. fumigatus* and *A. niger*, which underscores the potential of this approach.

## Figures and Tables

**Figure 1 jof-07-00270-f001:**
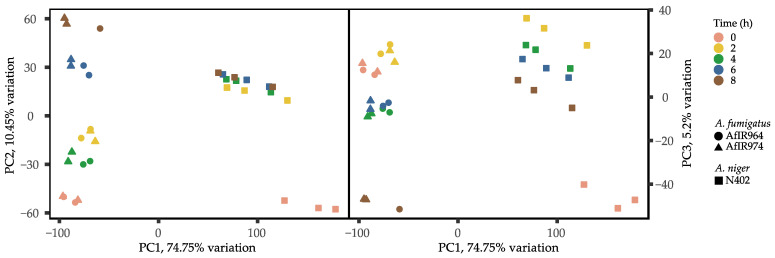
Principal component analysis of the log transformed expression data. The left plot shows PC1 on the *x*-axis and PC2 on the *y*-axis. The right plot shows PC1 on the *x*-axis and PC3 on the *y*-axis.

**Figure 2 jof-07-00270-f002:**
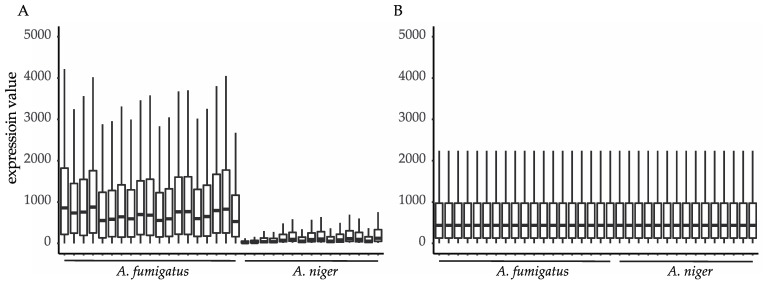
Expression profiles of the *A. fumigatus* and *A. niger* datasets before normalization (**A**) and after normalization **(B**). Outliers were left out of the figure for better visualization of the data.

**Figure 3 jof-07-00270-f003:**
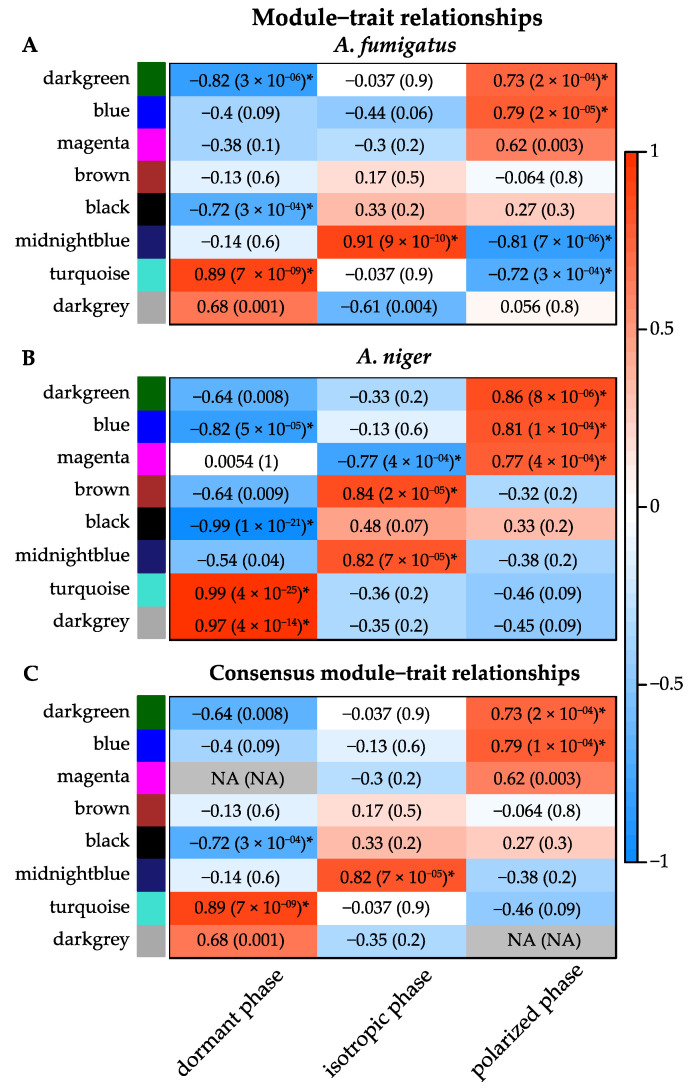
Module–trait relationships (**A**,**B**). Correlation of the module eigengenes (row) for *A. fumigatus* and *A. niger* with the external trait (column). (**C**) Correlation of the consensus modules with the external trait: A and B are summarized into one, where the lowest correlation score was used if the two correlations had the same sign, and zero relationships if the two correlations had opposite signs (NA). The module eigengenes correlation and *p*-value to the external trait are indicated in the cells and colored by the strength of the correlation. Red is a strong positive correlation compared to the dormant phase, while blue is a strong negative correlation compared to the dormant phase. * Significant modules have a correlation ≥ |0.70| and *p*-value ≤ 0.01. Only modules that have a significant correlation with an external trait are shown in this figure.

**Figure 4 jof-07-00270-f004:**
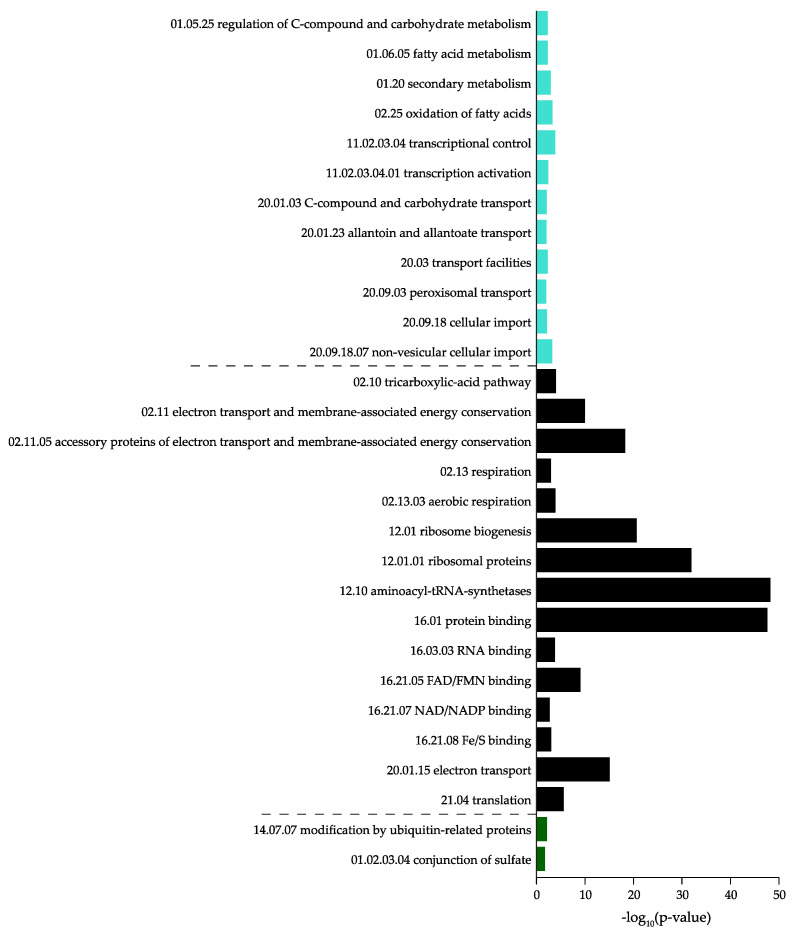
Functional categorization of the *turquoise, black,* and *darkgreen* modules.

**Figure 5 jof-07-00270-f005:**
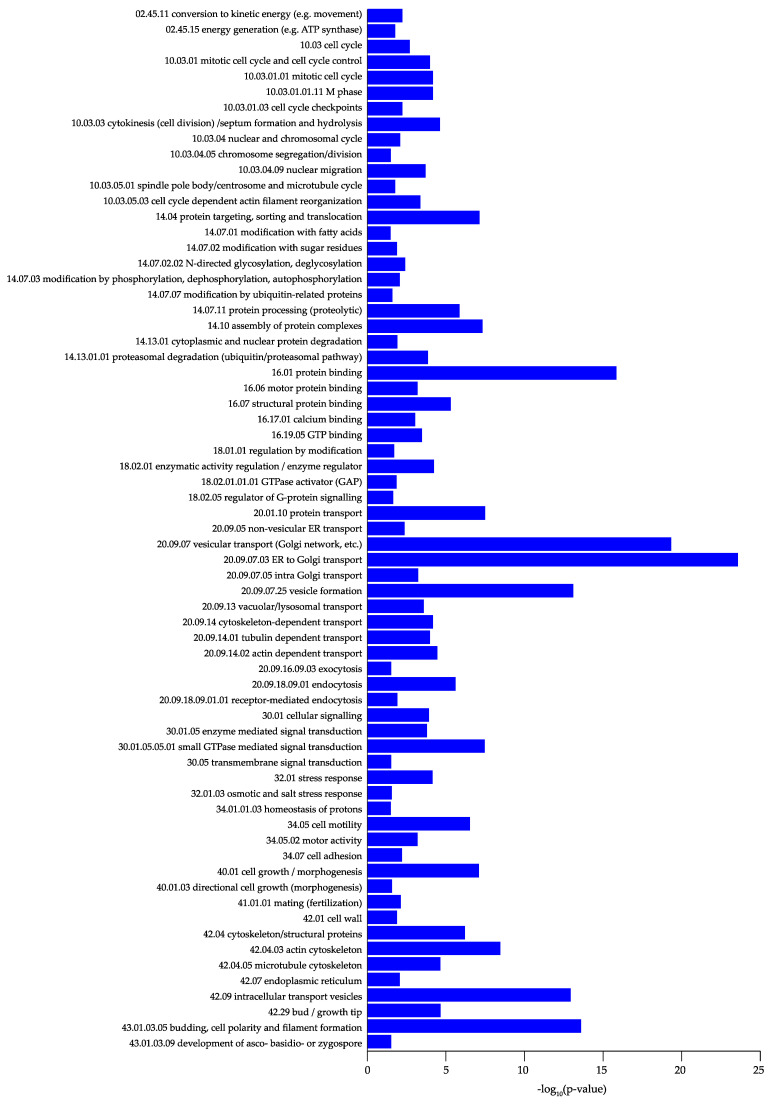
Functional categorization of the *blue* module.

**Figure 6 jof-07-00270-f006:**
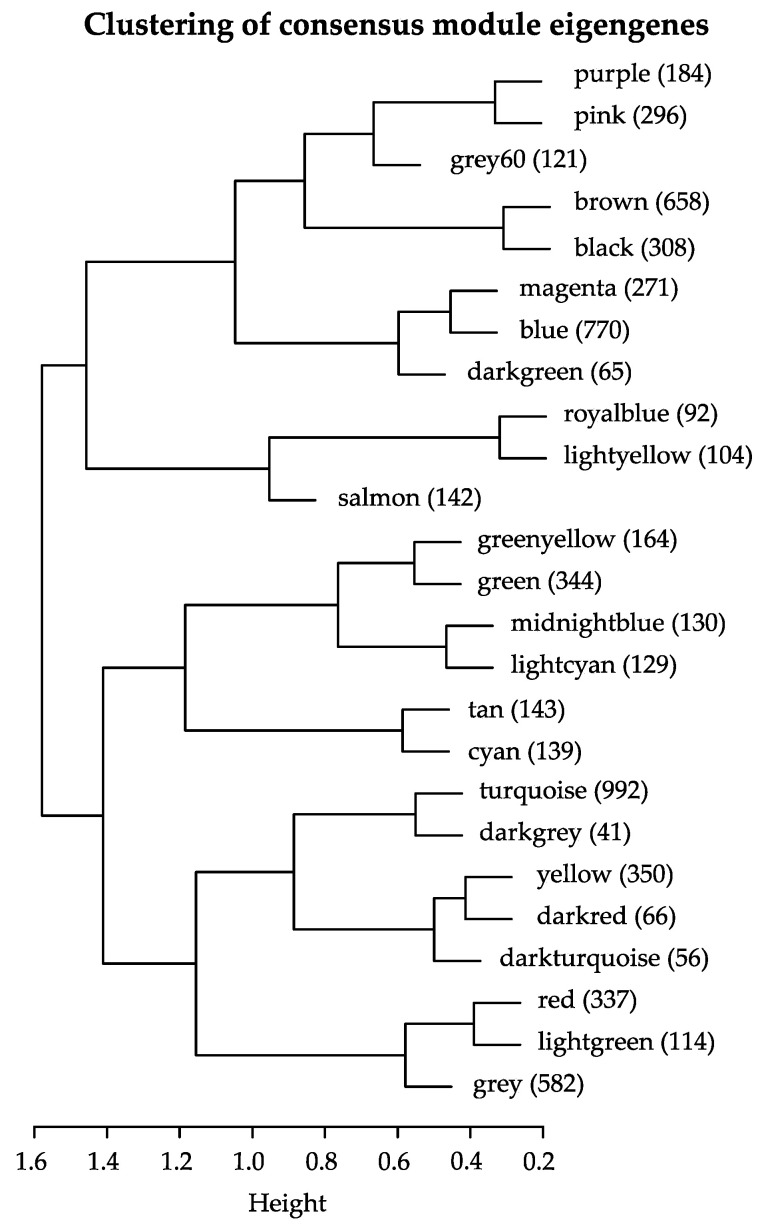
Clustering dendrogram of consensus module eigengenes for identifying meta-modules. Modules were clustered using the first principal component to form a dendrogram of consensus module eigengenes, representing the relation in expression between the consensus module eigengenes. The number of genes in each module is indicated between parentheses.

**Figure 7 jof-07-00270-f007:**
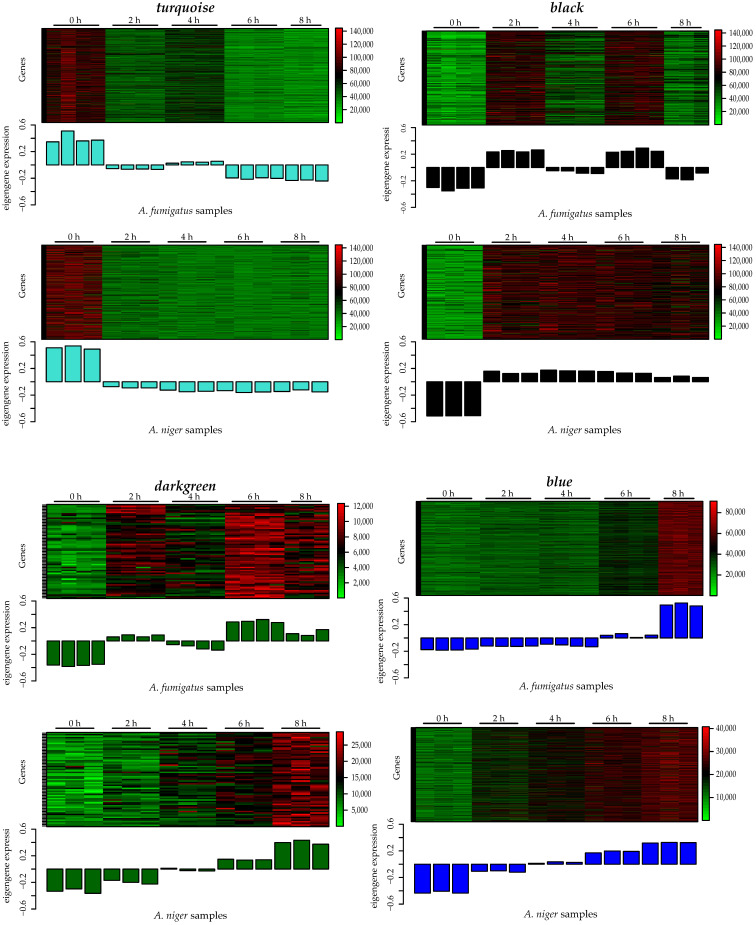
**A**–**D**. Heatmap of the gene expressions in each module (top) and expression levels of the corresponding module eigengene across the samples (bottom) for *A. fumigatus* and *A. niger.* The color bar shows the expression range of each module from the lowest to highest expression value.

**Table 1 jof-07-00270-t001:** Genes involved in biological processes associated with polarized growth.

Biological Process	*A. niger*	*A. fumigatus*	Description
Secretion-related GTPases and interacting proteins	*sarA*/An01g04040	Afu1g04940	Small GTPase of the ARF family, likely a component of COPII coat of transport vesicles
	An01g06060	*rab11*/Afu1g02190	Ras GTPase
	An08g03690	Afu1g11730	Ortholog(s) have GTPase activity and role in Golgi to plasma membrane transport
	*srgA*/An14g00010	*srgA*/Afu4g04810	Putative Rab GTPase
	An02g07780	Afu3g12080	Ortholog(s) have GTPase activity, enzyme activator activity, mRNA binding activity
	*geaA*/An18g02490	Afu5g11900	Putative guanine nucleotide exchange factor
Exocyst	An08g05570	Afu1g12790	Ortholog(s) have a role in golgi to plasma membrane transport, endoplasmic reticulum inheritance, establishment or maintenance of cell polarity, exocyst assembly
	An08g07370	Afu6g11370	Ortholog(s) have role in golgi to plasma membrane transport, exocyst assembly, exocyst localization
	An14g00010	*srgA*/Afu4g04810	Putative Rab GTPase
	*cftA*/An02g14200	*cdc42*/Afu2g05740	Rho GTPase
	*rhoA*/An18g05980	*rho1*/Afu6g06900	Ras GTPase
SNAREs and SNARE interactions	An02g01580	*sec17*/Afu2g12870	Putative vesicular fusion protein
	An04g07020	Afu4g10040	Ortholog(s) have golgi cisterna, endosome localization
	An07g02170	Afu7g05735	Putative v-SNARE
	An07g09960	Afu1g07420	Putative v-SNARE
	An15g01380	Afu6g04150	Ortholog(s) have SNAP receptor activity and a role in ER to golgi vesicle-mediated transport, retrograde vesicle-mediated transport, golgi to ER, vesicle fusion with golgi apparatus
Cell wall biosynthesis	*cfcD*/An01g05360	Afu1g02800	Putative chitinase
	An03g06220	*gel5/*Afu8g02130	Putative 1,3-beta-glucanosyltransferase
	An05g00130	Afu2g07590	Ortholog(s) have role in (1->6)-beta-D-glucan biosynthetic process
	*crhB*/An07g07530	*crh2*/Afu2g03120	Putative cell wall glucanase
	An08g07350	*gel2*/Afu6g11390	GPI-anchored 1,3-beta-glucanosyltransferase
	*chsD*/An09g02290	*chsF*/Afu8g05630	Chitin synthase
	*agtA*/An09g03100	*amyA*/Afu3g00900	Alpha-amylase
	An16g02850	*crh3*/Afu3g09250	Putative beta-glucanase
	An16g07040	*btgE*/Afu8g05610	Putative cell wall glucanase
Cell end markers	An18g04780	Afu1g06090	Ortholog(s) have a role in apical protein localization, regulation of cell shape and cell septum, plasma membrane of cell tip, spitzenkorper localization
Cdc42 complex	*cftA*/An02g14200	*modA*/Afu2g05740	Rho GTPase
	*cdc24*/An04g05150	*cdc24*/Afu4g11450	Ortholog(s) have Rho guanyl-nucleotide exchange factor activity
	*racA/*An11g10030	*racA*/Afu3g06300	Rho GTPase involved in regulation of cell polarity
Polarisome	*spaA*/An07g08290	Afu2g03710	Ortholog(s) have role in establishment of cell polarity, establishment, or maintenance of cell polarity, hyphal growth and hyphal tip polarisome localization
Arp 2/3 complex	An08g06400	Afu1g13330	Ortholog(s) have actin filament binding activity, role in Arp2/3 complex-mediated actin nucleation, endocytosis, spore germination and Arp2/3 protein complex, actin cortical patch localization
	An01g05510	Afu1g02670	Ortholog(s) have actin filament binding activity, role in Arp2/3 complex-mediated actin nucleation and Arp2/3 protein complex localization
	An12g08380	Afu6g02370	Ortholog(s) have actin filament binding, molecular adaptor activity, role in Arp2/3 complex-mediated actin nucleation, actin cortical patch organization, spore germination and Arp2/3 protein complex localization
	An16g01570	Afu5g01860	Ortholog(s) have actin filament binding activity

## Data Availability

The raw data were accessed through the NCBI Gene Expression Omnibus accession number GSE36439 (www.ncbi.nlm.nih.gov/geo/, accessed on 24 March 2020) and the NCBI Sequence Read Archive accession number PRJNA408076 (https://www.ncbi.nlm.nih.gov/sra, accessed on 24 March 2020).

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
