# Peer review of "Identifying Conserved Generic Aspergillus spp. Co-Expressed Gene Modules Associated with Germination Using Cross-Platform and Cross-Species Transcriptomics"

_jof, 2021, doi:10.3390/jof7040270_

Round 1

Reviewer 1 Report

In this manuscript Baltussen et al assess two previously generated transcriptomic datasets; a microarray from Aspergillus niger and RNA-seq dataset from Aspergillus fumigatus for common expression profiles via consensusWGCNA. Individual modules undergo functional enrichment analysis to find common biological functions during germination. This manuscript will be of interest to the field. However, several points need addressing and would require extra work to explore the biological relevance of presented analysis. 

Major points:

From the original studies A. fumigatus AfIR974 + AfIR964 and A. niger N402 were used. On CADRE the file that is mentioned in the Materials & Methods is neither in the folder for Af293 or A1163, which protein sequences have been used and why this one? Which is the closest related to the strains used? Additionally, why is the CBS 513.88 protein sequence file used and not the appropriate N402. Within species there is a lot of heterogeneity, for A. niger lineages see: https://doi.org/10.1007/s00449-020-02347-z and for A. fumigatus see: https://doi.org/10.1093/mmy/myaa075

Table S1 is not included in the files, so can't comment on this.

How did the authors come to 6,598 orthologous genes? de Vries et al 2017 (10.1186/s13059-017-1151-0) show ~8,500 genes that are panfungal and an additional 1000 specific to Aspergilli. 6,598 seems awfully low. Have the cut-offs been explored enough?

The original paper with the RNA-seq from A. fumigatus shows 20 datasets, why are there only 19 presented here?

The PCAs of A. fumigatus and A. niger need to be combined in one plot to make a clear statement about variation between all the samples. Additionally, principal components can be driven by outlier values, which are present in these datasets. These need to be trimmed.

Figure 1: The data after normalisation without the outliers removed needs to be shown as well, or alternatively these outlier values need to be completely removed from the dataset and further analysis. How many values are outliers per condition?

It seems from the PCA plot on Aspergillus niger that the variation between samples is explained on PC2, which is only 7.21% variation. Does this mean the microarray data is not so different between samples. 

PCA plots are not described in the results section. Merely, mentioned that the effect is visible. Explain what we are looking at.

What is not sufficiently addressed is how to differentiate between difference/similarity observed due to assay-to-assay variability, difference due to different experimental setup (pre culture was different and affects germination: https://doi.org/10.1101/797076 growth medium was different which might actually be the main driver of differences and similarities observed, and RNA extraction + library prep was different) and actual biological difference/similarity.

The turquoise module represents secondary metabolism. Have the authors looked into what secondary metabolites gene clusters this represents. Secondary metabolism is completely different between A. fumigatus and A. niger (https://doi.org/10.1073/pnas.1715954115) so what genes are linked here? Similar problem with not using the correct reference protein database for the transcriptomic data. There is a major difference in secondary metabolism between A. niger strains (https://doi.org/10.1186/1471-2180-13-91) and A. fumigatus strains (https://doi.org/10.1534/genetics.120.303549)

Table 1-4 need further annotation; FungiFun puts out P-values and how many genes are in each FunCat (+ how many genes this category comprised of) - this data needs to be added to fully understand this data. This data is usually put out as a plot instead of a table which makes it easier to interpret. A good example is: 10.1128/mBio.02962-19

Figure 4: would be easier to flip the dendrogram to make it legible. Could be improved by adding information of the size of each module.

Description of each module is just listing what categories are in each module. This can be read from tables, so feels redundant. Needs improving to frame it within the morphological context of germination.

Midnight blue is the only category correlating with isotropic growth, but does not have any FunCats associated. This means isotropic growth can't be characterised? Have the authors just looked at what genes are in this module and if they can make sense of it?

Figure 5 is not consistent; Eigengene expression Y-axis are different and colour intensity range is not described. Red is over-expression and green under-expression, compared to what?

Conclusion last paragraph - it needs to be mentioned that antifungals targeting germination genes would only work as prophylaxis. Normally, only established infections consisting of hyphae will be diagnosed and treated. Furthermore, not all genes important for germination will be good anti fungal targets. What makes a good target for prophylaxis needs to be described in more detail. How many genes are of interest for further detailed analysis?

The title mentions "germination pathways". However, no pathways are uncovered and described. Modules with similar biological function are found. Title needs to changed to reflect this.

Line 325-417: I don't think it flows well as a story, it needs some reshuffling and link to stages of germination.

Minor points:

Line 36; missing a full stop after oceans

Line 37: "evolved into an (opportunistic) pathogenicity" is not the correct choice of words here, needs revision

Line 40: "conidia of several fungal genera". Do the authors mean conidia of fungal species in different genera? It is odd choice of words.

Line 40-41: "are present in air all over the planet [3]". Not really addressed in ref 3 and not sure if there is air sampling data from all over the globe available. Definitely not on oceans, polar regions.

Line 44: "grow at body temperatures" is not the only fungal factor required, many of the Aspergilli grow at 37 Celsius

Line 46: "this has led to severe lung disease" should be "this can lead.."

Line 50: "if diagnosis is delayed or missed" would add infection with antifungal resistance here, which increases mortality rates significantly as well

Line 54-55: Epithelial cells have a significant role in clearance of inhaled conidia. Reviewed in https://doi.org/10.3390/jof4010008 

Line 57: "dormant conidia may not be pathogenic" author probably means, do not invoke an inflammatory response?

Line 63: "activation of dormant Aspergillus conidia". Don't think activation is the correct term here. Breaking of dormancy maybe?

Line 63-67: The authors might have missed a recent publication on germination, the nutritional requirements for breaking of dormancy and analysis of processes required for each stage of germination (https://doi.org/10.3390/jof7010030)

Line 156: "the biological meaning" needs rewording

Line 159 - 160: Not clear what is meant with the last line, rephrase.

Line 164: Illumina RNA-seq is not a profiling platform. Illumina supplies several platforms (MiSeq, HiSeq, NovaSeq).

Line 168-170: This phrase should be moved to materials and methods - is actually already mentioned there in the same way.

Line 175-177: "the dormant conidia were substantially different from all other time points in that the first principal component was responsible for the variation". The PC is not responsible for the variation on a sample. It shows the variation between samples. 

Line 177-179: Same comment as previous.

Line 181 (Figure 1): A. fumigatus needs to be italic

Line 190-192: Genes form a functional group, not a pathway per se. These two lines basically repeat the previous line, so can be removed.

Line 226: "biological meaning" = biological function

Line 327-330 there is A. fumigatus and A. niger data on this which is not mentioned.

Line 352-353: A. fumigatus gene IDs need to be mentioned too.

Line 361-362 same as previous comment

Line 357: exocyst was originally identified in 1996 (PMID: 8978675)

Line 370-371: Mention A. fumigatus IDs

Line 374-375: Same as above

Line 373-375: Cell wall biosynthesis genes is a broad term. There are over 100 cell wall biosynthesis genes. Genes need to be put into pathways (https://doi.org/10.1146/annurev-micro-030117-020406 https://doi.org/10.1016/j.fgb.2010.12.007 PMID: 11800273 and a lot of JP Large work)

Line 442: This line needs to be rephrased. Also, the taxonomy of A. niger within the genus has been recently changed (10.1128/mBio.00925-19)

Line 456-458; Also been shown by RNA-seq data from Danion et al (https://doi.org/10.3390/jof7010030)

Line 518: Arthur Ram

Line 552: This reference is not shown correct

Author Response

Dear Reviewer,

Please find our revised manuscript jof-1137817 entitled “Identifying conserved generic Aspergillus spp co-expressed gene modules associated with germination using cross-platform and cross-species transcriptomics”. Thank you for critically reviewing our manuscript. We have carefully taken all comments under consideration and made the appropriate revisions as indicated point by point in the response to reviews document. We believe that the comments made by the reviewers have improved our manuscript. We hope that our rebuttal sufficiently addressed all of the issues raised by the reviewers.

Sincerely yours,

Tim Baltussen

Note: Line numbers mentioned in the rebuttal refer to the line numbers in the marked-up version of the revised manuscript.

From the original studies A. fumigatus AfIR974 + AfIR964 and A. niger N402 were used. On CADRE the file that is mentioned in the Materials & Methods is neither in the folder for Af293 or A1163, which protein sequences have been used and why this one? Which is the closest related to the strains used? Additionally, why is the CBS 513.88 protein sequence file used and not the appropriate N402. Within species there is a lot of heterogeneity, for A. niger lineages see: https://doi.org/10.1007/s00449-020-02347-z and for A. fumigatus see: https://doi.org/10.1093/mmy/myaa075

For the RNASeq study (Baltussen et al. 2018), the illumina reads were mapped to the genome sequence of A. fumigatus af293 CADRE 30 (release 30), accordingly, the protein sequences of af293 CADRE release 30 were used in this study. The fasta files of this version can be downloaded from http://ftp.ebi.ac.uk/ensemblgenomes/pub/release-30/fungi/fasta/aspergillus_fumigatus/. The link to af293 release 30 was added to the Materials and Methods. Lines 112 - 113

The microarray chip used in the A. niger transcriptome study (van Leeuwen et al. 2013) was designed using the genome sequence of A. niger CBS513.88, accordingly, the protein sequences of this strain were used. https://doi.org/10.1038/nbt1282

Table S1 is not included in the files, so can't comment on this.

Table S1 is added to the supplementary files.

How did the authors come to 6,598 orthologous genes? de Vries et al 2017 (10.1186/s13059-017-1151-0) show ~8,500 genes that are panfungal and an additional 1000 specific to Aspergilli. 6,598 seems awfully low. Have the cut-offs been explored enough?

In order to analyse and compare the two datasets, they needed to be integrated one to one based on orthologous genes. We chose to use the RBH method as this method is very useful when only the best orthologous gene pairs between RBH method. Many studies apply an e-value cut-off of £1x10e-5 or £1x10e-6 in combination with a coverage of ³50% of any of the protein sequences in the alignment https://doi.org/10.1093/bioinformatics/btm585, https://doi.org/10.1371/journal.pone.0101850, https://doi.org/10.1002/0471250953.bi0301s42. OrthoMCL also uses an e-value cut-off of £1x10-5 http://www.genome.org/cgi/doi/10.1101/gr.1224503. In our study we used an e-value cut-off of 5x10-6 and a minimum coverage of 60%. Additionally, a bit score cut-off of ³ 80 was used, which is usually set between 50 – 80 https://doi.org/10.1002/0471250953.bi0301s42. The orthology tool Inparanoid uses a minimum bit score of 50 https://doi.org/10.1371/journal.pone.0000383.

Finding best hits involved sorting of the results from lowest to highest e-value, then, from highest to lowest bit-score. The first hit was therefore the best hit, however, the next hit with the same score would also be a best hit (co-orthologs). In our study, we only selected the first best hits as the datasets had to be integrated one to one based on orthologous gene pairs. This may, partly, be the cause of the difference in identified orthologous genes between de Vries (2017) and our study. Additionally, two different methods were used which could also cause a difference in identified orthologous genes. For example, the orthoMCL clustering algorithm has a parameter which is important for the clustering tightness.

Added to discussion. Lines 826 – 834

The PCAs of A. fumigatus and A. niger need to be combined in one plot to make a clear statement about variation between all the samples. Additionally, principal components can be driven by outlier values, which are present in these datasets. These need to be trimmed.

We agree with the reviewer that the expression data of both A. fumigatus and A. niger need to be combined in one plot. Outliers were not trimmed from the dataset, nonetheless, to avoid that the variance measure is dominated by highly expressed, highly variable genes, we log transformed the data using the function log1p() which uses the natural logarithm on 1+x.

Material and methods 2.3. Lines 133 – 137

Results 3.1. Lines 194 – 214

New PCA plot -> Figure 1

Figure 1: The data after normalisation without the outliers removed needs to be shown as well, or alternatively these outlier values need to be completely removed from the dataset and further analysis. How many values are outliers per condition?

We agree with the reviewer that the normalized data with outliers needs to be shown as well for transparent presentation of the data. This was plotted in Figure S1. Lines 219 – 220.

We don’t agree with the reviewer that the outlier values need to be completely removed from the dataset for downstream analyses as RNA-Seq data has a large dynamic range which includes outliers doi:10.1186/s13059-014-0550-8. The outliers are part of the natural variation and capture valuable information of the study. Excluding these values due to their extremeness can affect the results by removing information about the variability.

It seems from the PCA plot on Aspergillus niger that the variation between samples is explained on PC2, which is only 7.21% variation. Does this mean the microarray data is not so different between samples.

The PCAs of A. niger and A. fumigatus are now combined in one plot. Variation between samples on PC2 is small. PC3 was also plotted in the same figure which shows variation between A. niger samples. The combined PCA plot shows that PC1 explains species variation and the other PCs explain sample variation. Lines 194 – 214.

PCA plots are not described in the results section. Merely, mentioned that the effect is visible. Explain what we are looking at.

See previous comment. Lines 194 - 214

What is not sufficiently addressed is how to differentiate between difference/similarity observed due to assay-to-assay variability, difference due to different experimental setup (pre culture was different and affects germination: https://doi.org/10.1101/797076 growth medium was different which might actually be the main driver of differences and similarities observed, and RNA extraction + library prep was different) and actual biological difference/similarity.

Discussion on the assay-to-assay differences was added. Lines: 852 – 870

The turquoise module represents secondary metabolism. Have the authors looked into what secondary metabolites gene clusters this represents. Secondary metabolism is completely different between A. fumigatus and A. niger (https://doi.org/10.1073/pnas.1715954115) so what genes are linked here? Similar problem with not using the correct reference protein database for the transcriptomic data. There is a major difference in secondary metabolism between A. niger strains (https://doi.org/10.1186/1471-2180-13-91) and A. fumigatus strains (https://doi.org/10.1534/genetics.120.303549)

The FunCat category Secondary metabolism contained 117 genes. The webtool SMURF was used with the A. fumigatus IDs and chromosome localisation information as input. However, no clusters were found in the output, only a small number of backbone genes. The same was done using the A. niger IDs with similar results. Here we face the problem of not having the correct protein references available for the strains used in the studies. SMURF uses the start and stop location of each protein to search for clusters. However, these clusters could have different locations in different strains which makes it difficult to perform this analysis.

Table 1-4 need further annotation; FungiFun puts out P-values and how many genes are in each FunCat (+ how many genes this category comprised of) - this data needs to be added to fully understand this data. This data is usually put out as a plot instead of a table which makes it easier to interpret. A good example is: 10.1128/mBio.02962-19

Table 1-4 were removed from the manuscript. These results are now shown using horizontal bar plots. See Figure 6 and 7.

Figure 4: would be easier to flip the dendrogram to make it legible. Could be improved by adding information of the size of each module.

See Figure 4.

Description of each module is just listing what categories are in each module. This can be read from tables, so feels redundant. Needs improving to frame it within the morphological context of germination.

Listing of the categories in the manuscript was reduced. Table S2 shows the complete FungiFun2 output. FunCat categories were placed within context of germination.

Lines: 344 – 354

Lines: 419 – 428

Lines: 486 – 490

Lines: 521 – 523

Lines: 580 – 581

Lines: 656 – 661

Midnight blue is the only category correlating with isotropic growth, but does not have any FunCats associated. This means isotropic growth can't be characterised? Have the authors just looked at what genes are in this module and if they can make sense of it?

The module contained ~40 genes without annotation and almost all other annotations were putative. Among the annotated genes were some synthases (Siroheme, putative glucan) dehydrogenases (putative alcohol, proline) and decarboxylases (benzoylformate and glutamate).

The midnightblue module was discussed. Lines 883 - 889.

Figure 5 is not consistent; Eigengene expression Y-axis are different and colour intensity range is not described. Red is over-expression and green under-expression, compared to what?

Figure 5 adjusted. Y-axis identical in each plot and colour bars were added. Line 476

Conclusion last paragraph - it needs to be mentioned that antifungals targeting germination genes would only work as prophylaxis. Normally, only established infections consisting of hyphae will be diagnosed and treated. Furthermore, not all genes important for germination will be good anti fungal targets. What makes a good target for prophylaxis needs to be described in more detail. How many genes are of interest for further detailed analysis?

A short description of good anti fungal targets and prophylaxis was added. Lines 917 – 921.

The title mentions "germination pathways". However, no pathways are uncovered and described. Modules with similar biological function are found. Title needs to changed to reflect this.

Title changed to ‘Identifying conserved generic Aspergillus spp co-expressed gene modules associated with germination using cross-platform and cross-species transcriptomics’

Line 325-417: I don't think it flows well as a story, it needs some reshuffling and link to stages of germination.

The part was reorganised and some extra sentences and headings were added. Lines 418 – 661.

Minor points:

Line 36; missing a full stop after oceans

Full stop added. Line 36

Line 37: "evolved into an (opportunistic) pathogenicity" is not the correct choice of words here, needs revision

‘Evolves’ changed to ‘develops’. Line 36/37

Line 40: "conidia of several fungal genera". Do the authors mean conidia of fungal species in different genera? It is odd choice of words.

‘conidia of several fungal genera’ changed to ‘conidia of fungal species in different genera’. Line 40

Line 40-41: "are present in air all over the planet [3]". Not really addressed in ref 3 and not sure if there is air sampling data from all over the globe available. Definitely not on oceans, polar regions.

Sentence slightly changed and a new reference added. Line 40/41

Line 44: "grow at body temperatures" is not the only fungal factor required, many of the Aspergilli grow at 37 Celsius

‘Hypoxic conditions’ added as required factor. Line 51

Line 46: "this has led to severe lung disease" should be "this can lead.."

‘has led’ changed to ‘can lead’. Line 53

Line 50: "if diagnosis is delayed or missed" would add infection with antifungal resistance here, which increases mortality rates significantly as well

‘or if infection is caused by an antifungal resistant strain’ added. Lines 58 and 59

Line 54-55: Epithelial cells have a significant role in clearance of inhaled conidia. Reviewed in https://doi.org/10.3390/jof4010008

The role of epithelial cells in conidia clearance added. Line 63

Line 57: "dormant conidia may not be pathogenic" author probably means, do not invoke an inflammatory response?

‘not be pathogenic’ changed to ‘be immunologically inert’. Line 66

Line 63: "activation of dormant Aspergillus conidia". Don't think activation is the correct term here. Breaking of dormancy maybe?

‘activation of dormant’ changed to ‘Breaking of dormancy in’. Line 72

Line 63-67: The authors might have missed a recent publication on germination, the nutritional requirements for breaking of dormancy and analysis of processes required for each stage of germination (https://doi.org/10.3390/jof7010030)

Publication added. Line 74

Line 156: "the biological meaning" needs rewording

‘biological meaning’ changed to ‘biological function’. Line 182

Line 159 - 160: Not clear what is meant with the last line, rephrase.

Rephrased, see lines 184 and 186

Line 164: Illumina RNA-seq is not a profiling platform. Illumina supplies several platforms (MiSeq, HiSeq, NovaSeq).

‘RNA-Seq’ changed to ‘NextSeq500’. Line 190

Line 168-170: This phrase should be moved to materials and methods - is actually already mentioned there in the same way.

Phrase removed from Results section.

Line 175-177: "the dormant conidia were substantially different from all other time points in that the first principal component was responsible for the variation". The PC is not responsible for the variation on a sample. It shows the variation between samples.

See lines 194 - 214

Line 177-179: Same comment as previous.

See previous comment.

Line 181 (Figure 1): A. fumigatus needs to be italic

Changed. Figure 2 in revised manuscript

Line 190-192: Genes form a functional group, not a pathway per se. These two lines basically repeat the previous line, so can be removed.

Phrase removed.

Line 226: "biological meaning" = biological function

‘biological meaning’ changed to ‘biological function’. Line 309

Line 327-330 there is A. fumigatus and A. niger data on this which is not mentioned.

Not sure what data the reviewer refers to here.

Line 352-353: A. fumigatus gene IDs need to be mentioned too.

  1. fumigatus IDs added.

Line 361-362 same as previous comment

  1. fumigatus IDs added.

Line 357: exocyst was originally identified in 1996 (PMID: 8978675)

Original publication added. Line 583

Line 370-371: Mention A. fumigatus IDs

IDs added.

Line 374-375: Same as above

IDs added.

Line 373-375: Cell wall biosynthesis genes is a broad term. There are over 100 cell wall biosynthesis genes. Genes need to be put into pathways (https://doi.org/10.1146/annurev-micro-030117-020406 https://doi.org/10.1016/j.fgb.2010.12.007 PMID: 11800273 and a lot of JP Large work)

CAZy families were used to classify cell wall biosynthesis genes. Lines 547 - 554

Line 442: This line needs to be rephrased. Also, the taxonomy of A. niger within the genus has been recently changed (10.1128/mBio.00925-19)

Rephrased and publication discussed. Lines 846 - 851

Line 456-458; Also been shown by RNA-seq data from Danion et al (https://doi.org/10.3390/jof7010030)

Reference added. Lines 852 – 855.

Line 518: Arthur Ram

Thank you for noticing.

Line 552: This reference is not shown correct

Reference corrected.

Reviewer 2 Report

Broad comments

This work by Baltussen et al. describes the analysis of transcriptomic data obtained by microarray and RNAseq from two species of human pathogenic Aspergillus. The objective of this study is to find biological processes that are essential in the process of germination, and consequently, that may be important for the ability of fungal pathogens to survive and start growth inside the host. The genes expression data belongs to previous studies and corresponds to multiple time points that include the transition from resting conidia to isotropic and polarized growth. First, the authors identified the orthologous pairs between A. fumigatus and A. niger. Next, they constructed a consensus gene co-expression network and, using WGCNA, 25 gene modules were detected and associated to biological processes using enrichment analysis. Four of these modules were associated with the process of germination.

The authors describe the modules containing significant gene expression patterns in each stage and they find, mostly, genes and routes that have been already studied or related to polarity establishment in fungi. It is not clear whether new pathways are found in this study. This should be explained in more detail.

In addition, the discussion section is poor. Maybe results and discussion can be rearranged.

Specific comments

36: …oceans.

124: generate

361: homologs

424: …fumigatus, enabling…

459: …[21]. The…

470: …[20], whereas…

470: transcripts

Author Response

Note: Line numbers mentioned in the rebuttal refer to the line numbers in the marked-up version of the revised manuscript.

The authors describe the modules containing significant gene expression patterns in each stage and they find, mostly, genes and routes that have been already studied or related to polarity establishment in fungi. It is not clear whether new pathways are found in this study. This should be explained in more detail.

The manuscript title was adjusted to reflect the results and conclusion better.

In addition, the discussion section is poor. Maybe results and discussion can be rearranged.

Discussion section adjusted. See lines 826 - 834, 846 – 870, 883 – 889, 893 – 895, 917 – 921.

Specific comments

36: …oceans.

Full stop added. Line 36

124: generate

Changed. Line 136

361: homologs

Changed. Line 586.

424: …fumigatus, enabling…

Kept the full stop, otherwise the sentence would become very long.

459: …[21]. The…

Full stop added. Line 895

470: …[20], whereas…

Full stop changed to comma. 906

470: transcripts

‘Transcript’ changed to ‘transcripts’. Line 906

Round 2

Reviewer 1 Report

I would like to commend the authors on the extensive revisions and improvement of the analysis and manuscript as a whole. The additional information and figures make understanding the data and relevance much easier and improve the conclusions that can be drawn. I recommend accepting the manuscript in its current form and am sure it would be of wide interest to the readership.